# Modelling ATR-FTIR Spectra of Dental Bonding Systems to Investigate Composition and Polymerisation Kinetics

**DOI:** 10.3390/ma14040760

**Published:** 2021-02-05

**Authors:** António HS Delgado, Anne M. Young

**Affiliations:** 1Division of Biomaterials & Tissue Engineering, UCL Eastman Dental Institute, London NW3 2PF, UK; anne.young@ucl.ac.uk; 2Clinical Research Unit, Centro de Investigação Interdisciplinar Egas Moniz (CiiEM), Instituto Universitário Egas Moniz (IUEM), 2829-511 Caparica, Portugal

**Keywords:** dental adhesives, FTIR, infra-red spectroscopy, photopolymerisation, prediction model

## Abstract

Component ratios and kinetics are key to understanding and optimising novel formulations. This warrants investigation of valid methods. Attenuated Total Reflectance Fourier Transform Infra-Red (ATR)-FTIR spectra of separate primers/adhesives were modelled using summed spectra of solvents (water, ethanol), methacrylate monomers (HEMA (hydroxyethyl methacrylate), Bis-GMA (bisphenol A glycidyl methacrylate), and 10-MDP (10-methacryloyloxydecyl dihydrogen phosphate)), and fillers, multiplied by varying fractions. Filler loads were obtained following their separation from the adhesives, by analysing three repetitions (*n* = 3). Spectral changes during light exposure at 37 °C (20 s, LED 1100–1330 mW/cm^2^) were used to determine polymerisation kinetics (*n* = 3). Independent samples T-test was used for statistical analysis (significance level of 5%). FTIR modelling suggested a primer solvent percentage of OBFL (Optibond FL) (30%) was half that of CFSE (Clearfil SE 2) (60%). OBFL included ethanol and water, while CFSE included only water. Monomer peaks were largely those of HEMA with lower levels of phosphate monomers. OBFL/CFSE adhesive model spectra suggested that both contained equal volumes of Bis-GMA/HEMA, with CFSE having 10-MDP. Filler levels and spectra from OBFL (48 wt.%) and CFSE (5 wt.%) were different. Both systems reached a 50% conversion rate within seconds of light exposure. The final conversion for OBFL (74 ± 1%) was lower compared to CFSE (79 ± 2%) (*p* < 0.05). ATR-FTIR is a useful method to investigate relative levels of main components in bonding systems and their polymerisation kinetics. Such information is valuable to understanding such behaviour.

## 1. Introduction

Over the past several years, many new methacrylate monomer-based dental bonding systems have been formulated and commercialised for restorative purposes, with widely varying compositions and clinical capabilities. They provide a link between restorative material and enamel through micromechanical interlocking with etched hydroxyapatite prisms and by copolymerising with the composite methacrylate resin phase. Furthermore, they can help bond and seal dentine through the formation of a hybrid layer and resin tags within its tubules [1,2,3]. The hybrid layer is typically formed by polymerisation of monomers within a demineralised collagen network. In adhesive dentistry, it is persistently described as the weakest link [4,5]. The success of the dentine bond is determined by complex molecular interactions between the bonding system and dentine. They are greatly influenced by dentine variability and wettability, in addition to the bonding agent hydrophobicity, monomer functional groups and molecular weight, filler load, solvent, and acidity [6,7]. Unfortunately, the manufacturer’s information on the composition of bonding systems will include their main components but generally not their relative levels. This detailed information is paramount to understanding their behaviour and thereby improving clinical success.

Dental bonding systems have been grouped into various different categories. Two of the most successful, proven in long-term clinical and laboratory studies, that are the so-called gold-standard, are the 3 step Etch-and-Rinse (ER) and the 2-step Self-Etch (SE) systems Optibond FL (OBFL) and Clearfil SE 2 (CFSE) respectively [8,9,10]. Both of these employ a hydrophilic primer followed by the application of a more hydrophobic adhesive, with the exception that in OBFL, a separate phosphoric acid etching step is advocated.

The main components in primers usually include the hydrophilic monomer, hydroxyethyl methacrylate (HEMA) and solvents. Their hydrophilicity enables penetration of the water-rich organic matrix in dentine. Functional, adhesion promoting, monomers (e.g., 4-methacryloxyethyl trimellitate anhydride – 4-META, glycerophosphate dimethacrylate—GPDM and 10-methacryloyloxydecyl dihydrogen phosphate—10-MDP) are also included in contemporary systems [8,11]. These can provide self-etching characteristics, in addition to the possible ionic bonding of acidic carboxylic or phosphate groups to calcium from hydroxyapatite, respectively [12]. In the adhesive, hydrophobic, aromatic, crosslinking dimethacrylates such as bisphenol A glycidyl methacrylate (Bis-GMA) are included [3,13,14]. These make the adhesives readily compatible with the primer and composites, as Bis-GMA is often their primary base monomer. Furthermore, fillers are added to control flow and improve physical properties [1,8].

Following solvent evaporation, the monomers undergo a free-radical addition polymerisation reaction, giving rise to a set polymer. Each monomer system has a maximum optimal polymerisation level. A suboptimal polymerisation level and rate reduces physico-mechanical properties [10,15,16] and adhesive performance [12,17]. Low degree of conversion may provoke toxicity in pulp tissues, as free monomer elution is then possible, diffusing into the pulp complex and surrounding tissues. Thus, polymerisation rates and conversion in bonding systems are clinically relevant [15].

FTIR is widely used in studies of material characterisation and polymerisation kinetics [18,19]. It can quantify compositional changes in polymer chemistry, facilitating an understanding of the nature of the materials and how they may react when in contact with dental substrates [20,21]. FTIR has been extensively used to investigate bonding systems and, almost exclusively, to study final monomer conversions and the quality of the hybrid layer. Employing FTIR to assess their composition and track reaction rates has seldom been reported [22]. ATR-FTIR systems are particularly useful for the continuous monitoring of liquid and paste setting reactions. It has been employed to quantify the relative rates of polymerisation and acid base reactions in restorative materials [23,24]. Additionally, modelling methods such as difference spectra allow the study of peak changes owing to setting reactions [23].

The aim is therefore to develop a method to analyse the ATR-FTIR spectra of dental bonding systems using OBFL and CFSE as examples. This study provides a prediction model of the component ratios to better understand formulations from different manufacturers. To further characterise these systems, filler loads were determined and methods of quantifying chemical changes during polymerisation are provided.

## 2. Materials and Methods

### 2.1. Materials

The source and composition (according to the manufacturer) of the commercial bonding systems used in this study are listed in Table 1. The main monomer structures, molecular weight (g/mol), and aqueous solubility (Log (S mol/l)) obtained using ChemBioDraw Ultra, version 14.0; Software for chemical drawing (Perkin-Elmer, Waltham, MA, USA) are provided in Figure 1.

To model fit the spectra, the pure main components identified from manufacturer information were obtained. These included HEMA (DMG, Hamburg, Germany, code 11220), Bis-GMA (Polysciences, Warrington, PA, USA, code 03344), 10-MDP (DM Healthcare, San Diego, CA, USA, code P01030), and ethanol (Sigma-Aldrich, St. Louis, MO, USA). Additionally, triethyleneglycol dimethacrylate—TEGDMA (DMG, Hamburg, Germany, Code 100102) was also examined to provide the spectrum of a commonly used diluent monomer.

### 2.2. Filler Separation and Weight Percentage (Filler Load)

To separate the adhesive fillers from the monomers, 0.5 g of each adhesive was dissolved in 10 mL of acetone in a test tube of a known mass, agitated in a vortex mixer, and centrifuged for 15 min at 4000 rpm (Eppendorf 5804 R). Sediment mass was determined to the nearest 0.0001 g using a balance (Mettler AG204, Gemini, Tiel, The Netherlands) following the removal of the supernatant and overnight drying. This was reported as a weight percentage of the adhesive (*n* = 3).

### 2.3. FTIR Spectra of Components, Primers, Adhesives and Fillers

To obtain Fourier Transform Infra-Red (FTIR) spectra of primers, adhesives, fillers, monomers (HEMA, Bis-GMA, TEGDMA, 10-MDP) and solvents (water and ethanol), a diamond Attenuated Total Reflectance (ATR) accessory (Golden Gate ATR, Specac Ltd., Orpington, UK) in an FTIR spectrometer (Spectrum One, Perkin-Elmer, MA, USA) was employed. Spectral acquisition was almost immediate. For the solid adhesive fillers (not required for the liquids), the golden gate bridge of the ATR accessory was required to ensure good contact with its diamond. Timebase, version 3.1.4; Software for FTIR data analysis (Perkin-Elmer, Waltham, MA, USA) was used to calculate the ratio of the intensity obtained with versus without the sample, on the ATR diamond, and convert the data to absorbance versus wavenumber. In all cases spectra were acquired from 700 to 4000 cm^−1^ at a resolution of 4 cm^−1^.

### 2.4. Model for Estimation of Component Ratios

The FTIR spectra of key monomers used for modelling and assignments are provided in Figure 2 and Table 2. The common methacrylate group peaks and reference peaks are highlighted in Figure 2 by blue bands and arrows respectively and separated in Table 2.

Strong peaks due to solvents and common methacrylate peaks made it possible to assess relative levels of solvents versus methacrylates in primers with reasonable certainty. Conversely, the similarities in methacrylate spectra, lack of all possible control spectra, and peak shifts when pure components are mixed made it more difficult to identify specific monomers if present at a low level. Using strong, well separated aromatic, C–OH and phosphate peaks (see Table 2), did, however, enable some quantification of the relative levels of the main monomers with these reference groups.

To semi-quantify component ratios in OBFL and CFSE, a model was built using Microsoft Excel Tools, version 16.35 (Microsoft, King County, WA, USA) to fit the initial primer and adhesive FTIR spectra. The model assumed that the Beer–Lambert law could be applied [24], such that the absorbance of light (A_m,v_) at a given wavenumber (v) was equal to the sum of the absorbance due to each monomer and solvent multiplied by its volume fraction. For the model, the monomer spectra of Bis-GMA, HEMA, and 10-MDP were employed. These enabled an estimation of the relative levels of aromatic dimethacrylates (e.g., Bis-GMA) versus more hydrophylic hydroxyl (–OH) and carboxylic acid (–COOH) containing monomers (HEMA, MMEP, and GDMA) or phosphate containing monomers (MDP or GPDM) respectively. An additional term for the solid fillers and a flat background term B, possibly caused by scattering, had to be applied. The resulting Equation (1) was:(1)$$A_{m,v}=\;\sum_{r}(X_{r}\times A_{r,v})+\left(X_{f}\times A_{f,v}\right)+B.$$

Where model absorbance is *A_m_*, *X_r_* is the fraction of each pure solvent or monomer spectrum used in the model which for an ideal system would equal their volume fraction. *A_r_* is the absorbance of the monomer or solvent at each given wavenumber, v. *B* is a constant background absorbance, and *f* represents the filler phase which is added on top of the model as additional absorbance.

For peak fitting, the full spectral range was employed. The shape of the 3300 cm^−1^ region was used for determining relative levels of molecules containing –OH groups. The 1750–850 cm^−1^ region was useful for identifying relative levels of aromatic, hydrophilic, and phosphate containing monomers. To achieve the best fit, the component ratios were systematically varied, ensuring the first two rules in Equations (2)–(4) were obeyed. The best fit was determined once the modulus of the difference was at the minimum possible.
(2)$$\sum^{\;}\left(X_{r}\right)=1,$$
(3)$$\sum^{\;}\left(A_{p,v}-A_{m,v}\right)=0,$$
(4)$$\sum^{\;}(|A_{p,v}-A_{m,v}|)=minimum$$

Parameters used to achieve best fit: 2—the sum of the monomer and solvent (liquid) volume fractions was 1; 3—the sum of the difference between the actual and model spectra was equal to zero; and 4—the lower the sum of the modulus of the difference in the actual and model spectra, the better the fit. *A_p,v_* is the absorbance due to the primer or adhesive.

### 2.5. Polymerization Kinetics

To obtain changes in FTIR spectra of the combined primer and adhesive upon light curing, equal drops of primer and adhesive were confined within circlips of a 1 cm internal diameter and 1 mm depth (0.08 cm^3^ volume) placed around the ATR FTIR diamond. Two drops of the system (primer + adhesive) are typically 0.06 cm^3^ in volume, almost reaching 1 mm of depth. The ATR-FTIR system provides a level of polymerisation in the lower few micron depth of the sample. Firstly, the primer was added, after which evaporation of the solvent was carried out with a hairdryer, for 20 s, at temperature stage 1–1500 W (Grunding HD, 259, Nuremberg, Germany). The adhesive was then added, and an acetate sheet placed on top. The top surface of the material was irradiated with a single emission peak Light Emitting Diode (LED) Light Curing Unit (LCU) (Demi Plus, Kerr, Orange, CA, USA) in direct contact with the acetate. The power output ranged from 1100 mW/cm^2^ to 1330 mW/cm^2^, between 450 to 470 nm. FTIR spectra of the lower sample surface were obtained for 20 s before, then during and after 20 s of light exposure. Spectra were acquired from 700 to 4000 cm^−1^ at a resolution of 4 cm^−1^, for 20 min at 37 °C. The light curing began 20 s after the start of mixing and spectral acquisition. For spectral analysis, TimeBasem version 3.1.4 (Perkin-Elmer, Waltham, MA, USA) was used.
(5)$$D_{C}\;\left(\%\right)=100\times \left(h_{0}-h_{t}\right)/h_{0}$$
where *(h_0_)* and *(h_t_)* are the methacrylate C–O stretching peak absorbance at 1320 cm^−1^ above background at 1335 cm^−1^ initially and at time t after the start of polymerisation initiation, respectively. Obtaining spectra continuously during polymerisation without any disconnect from the ATR diamond enables the continuous monitoring of the exact same material volume during polymerisation. This removes the need for normalisation by a reference peak. The validity of this method has been verified previously in a wide range of studies [23,24]. The 1320 cm^−1^ peak was employed as an alternative to the more commonly employed methacrylate 1640 cm^−1^ C=C peak which cannot be used, since in the 1635 cm^−1^ region there is strong absorption due to water being present in the primers. From this data, the final degree of conversion and rate of polymerisation were determined. The maximum rate of polymerisation (R_pmax_), or reaction rate, was calculated using the first derivative of the *D_C_*% versus time curve and is shown in %/s.

### 2.6. Statistical Analysis

Microsoft Excel v16.35 (Microsoft, Redmond, WA, USA) was used for data modelling and descriptive statistics such as means and standard deviations. To test if final degrees of conversion values and rates of polymerisation were significantly different, SPSS v26.0 (IBM, Chicago, IL, USA) was used for hypothesis testing, by employing parametric independent samples T-test, to compare *D_C_* (%) means, and Mann–Whitney U for R_p,max_. The level of significance was set to 5%.

## 3. Results

### 3.1. Chemical Composition: OBFL and CFSE

#### 3.1.1. Primer Spectra

The FTIR spectra of the primers are given in Figure 3 and peak assignments in Table 3.

The large spectral differences between the two primers indicate major differences in solvent versus methacrylate levels. CFSE gives approximately double the amount of absorbance of OBFL in the 3300 cm^−1^ –OH peak region. Conversely, the 1640 cm^−1^ combined water and methacrylate C=C peak are of comparable height for both primers. All other main common methacrylate peaks between 900 and 1800 cm^−1^ are significantly lower for CFSE. Both primers have a strong OH peak at 1080 cm^−1^, relative to their other monomer contributions, consistent with ethanol, in OBFL, and/or HEMA in both. A shoulder at 1000 cm^−1^ is also observed on the methacrylate/ethanol 1050 cm^−1^ peak, which is suggestive of a phosphate containing monomer and/or filler absorbance.

#### 3.1.2. Adhesive Spectra

Compared with their primers, OBFL and CFSE adhesives showed greater similarities in their FTIR spectra as seen in Figure 4. Both adhesives give strong aromatic peaks, missing in the primer spectra, that are consistent with Bis-GMA (see Figure 4 and Table 3). Differences between the two adhesive spectra can be seen in the 1000–900 cm^−1^ region which may be attributable to differences in the filler and/or phosphate levels (Figure 4).

#### 3.1.3. Filler Spectra

FTIR spectra of the fillers are provided in Figure 5. For OBFL, a strong broad band is located at 992 cm^−1^. This may result from overlapping contributions of Si-O asymmetric bond stretching and the B-O bond vibration in the barium aluminoborosilicate glass. OBFL has a peak in the 1400 cm^−1^ region due to the B–O stretch. The band around 1200 cm^−1^ is related to Si–CH_2_, silanated particles, while the band around 1060–1080 cm^−1^ in both OBFL and CFSE are due to a Si–O asymmetric stretch of silica particles, and the band around 790 cm^−1^ is due to symmetric Si–O stretch in both systems.

#### 3.1.4. Model Spectra


**(A) Primers**


Fractional amounts of pure component spectra used for model construction are given in Table 4. According to the model, the OBFL primer contains 30% solvent (water and ethanol), while the CFSE primer is 60% water-based. It suggests both primers contain high levels of HEMA and lower levels of a phosphate monomer. The model gives a good fit for CFSE (Figure 6B) but is less good for OBFL (Figure 6A). This will be due to CFSE consisting primarily of components used in the model whilst OBFL has significant levels of additional monomers (MMEP and GPDM) not available for model fitting.

The difference spectrum for OBFL in Figure 6A, obtained by subtracting the model from the actual spectra, is consistent with the main additional monomer in OBFL being MMEP. This difference spectrum has peaks at 1270, 1640, and 1700 cm^−1^ which could be due to MMEP acidic (COO) and methacrylate (C=C and C=O) groups. The remaining 3 weak, sharp peaks between 1500 and 1600 cm^−1^ are similar to aromatic peaks in Bis-GMA (see Table 2) but slightly shifted in wavenumber. This would be consistent with the MMEP aromatic ring having COO groups attached. The imperfect agreement between the methacrylate model and actual peaks could additionally be due to the use of 10-MDP as a model for GPDM in OBFL. Comparing their chemical structures (Figure 1) suggests this would give greater model aliphatic C–H peaks (seen between 1350–1450 cm^−1^) but weaker methacrylate peaks (1600–1750 cm^−1^) than required and observed.


**(B) Adhesives**


Figure 7 illustrates model fitting for the adhesives of both bonding systems whilst spectrum fractions are provided in Table 4. This suggests equal levels of Bis-GMA and HEMA are in both adhesives. Other than Bis-GMA/HEMA, CFSE also contains 10% of the 10-MDP spectrum included in the model as shown in Figure 7B, marking its difference to OBFL, which did not contain phosphate monomers in the adhesive. Overlapping contributions in the 1200–750 cm^−1^ region owing to the Si-O stretch and B-O bonding vibrations are caused by fillers as shown in Figure 7. Slight differences between actual and model spectra in Figure 7B between 1240 and 940 cm^−1^ may be a consequence of solvent interactions with the 10-MDP ionising the phosphate group and the additional low level hydrophobic aliphatic dimethacrylate in Table 1.

### 3.2. Polymerisation Kinetics and Filler Load

Regarding filler content, OBFL had a mean average of 48 wt.%, while CFSE revealed a much lower weight percentage of filler (5 wt.%). The final extrapolated degree of conversion of CFSE (79 ± 2%) was higher than for OBFL (74 ± 1%) (Independent Samples T-Test, *p* = 0.04) whilst their overall reaction rates were similar (Mann–Whitney U, *p* > 0.05) (Table 5).

To monitor kinetics, and look at rates at earlier and later times, Figure 8 presents the *D_C_* (%) for the first 100 s.

The difference spectra obtained by subtracting the final from the initial spectra are given in Figure 9 for both mixed bonding systems. The methacrylate group peaks shift during the polymerisation process, particularly C=O, C=C, and C–O stretching vibrations leading to the observation of peaks and adjacent troughs. The level of change at each peak and trough is higher for CFSE compared with OBFL, consistent with a higher degree of conversion.

### 3.3. Agreement with the Information Supplied by the Manufacturers

In order to summarise the findings of this study and compare them to the information supplied by the manufacturers, the table below was made. The amount of information not available by manufacturers is clearly visible, and fractions did not always conform what was found in this study, although the fractions verified with FTIR are susceptible to deviations (Table 6).

## 4. Discussion

An ideal bonding system should ensure intimate adaptation, penetration, ionic interaction, and sealing with enamel and dentine to enable decent bond strengths. It should have good mechanical properties and a high degree of conversion to guarantee interfacial strength, be resistant to hydrolysis and dissolution, and also be biocompatible with oral tissues [7,25]. Depending on the bonding strategy, bonding systems exist in separate bottles or combined in one, and this, naturally, makes their compositions highly variable [2]. Chemical composition of dental adhesives will affect virtually all properties including viscosity, water sorption, polymerisation, physico-mechanical properties, and their ability to penetrate and bond. This influences the hybrid layer, formed upon in situ polymerisation [11,13,15], and its susceptibility to hydrolysis and degradation. The identification of the bonding system component ratios can help provide an understanding of all these properties.

OBFL and CFSE are considered gold-standard ER SE bonding systems respectively. This is a largely due to their observed longevity in class-V restorations, within clinical trials [10,26]. Regarding OBFL, it is consistently regarded as the bonding system with the highest microtensile bond strength and excellent retention. This may be due to its high filler content and elastic modulus, the pre-etching step which provides deep demineralisation and widens tubules and also the fact that it has two functional monomers [27,28]. CFSE is capable of a double bonding mechanism; firstly, the formation of micro-mechanical bonding through a 1 μm thick hybrid layer, and secondly it benefits from a strong bond between 10-MDP and hydroxyapatite and collagen [27].

Most commercial bonding systems contain HEMA to enhance miscibility between hydrophilic and hydrophobic components and to function as an adhesion promoter by improving permeability into dentine [29]. HEMA was found to be present at a high level in both primers and adhesives of OBFL and CFSE. HEMA-free bonding systems have been observed as more susceptible to phase separation and droplet formation, which produces easily degradable hybrid layers [30]. Too high amounts of HEMA, however, may also affect the integrity of the hybrid layer, owing to its hydrophilicity, leading to hydrolytic degradation [31,32]. This compromises adhesive and restoration longevity.

Bis-GMA, the main other monomer identified as present in OBFL and CFSE adhesives, has the highest monomer molecular weight compared to others in this study. Whilst Bis-GMA is practically insoluble in water due to its hydrophobic aromatic groups, HEMA, with the lowest molecular weight and a hydrophilic OH group is highly soluble [21]. GPDM and GDMA in the OBFL primer and adhesive respectively are moderately water-soluble due to their relatively small size and hydrophilic phosphate or OH groups respectively (see Figure 1). Conversely, MMEP and 10-MDP in OBFL and CFSE primers have limited aqueous solubility due to their aromatic groups and long aliphatic chains respectively. These therefore require the addition of co-solvents (HEMA or ethanol) to aid their dissolution in the water. Whilst HEMA, MEPP, and 10-MDP are monomethacrylates, GPDM, GDMA, and Bis-GMA all have two polymerizable methacrylate groups and are therefore crosslinkers.

The FTIR spectra in this study suggested CFSE had approximately 10% of the acidic-phosphate monomer 10-MDP in both the primer and adhesive. According to the literature, CFSE may have a higher concentration of this monomer than other bonding systems [33]. This molecule is a surfactant with a hydrophobic aliphatic tail and hydrophilic phosphate head group. It should therefore form various aggregate structures in solution and aligned layers at interfaces. These will be dependent upon a 10-MDP concentration and hydrophilicity of the solvent and surface. 10-MDP has been observed to form multiple nanolayers at the dentine interface particularly when at higher concentrations (>15 wt.%) [34]. This phenomenon has been reported to be limited in bonding systems that have HEMA, which is known to inhibit the effect [35].

In OBFL, a small peak (P–O stretch region at 1000 cm^−1^) was found in the primer FTIR spectrum, which is likely due to glycerophosphate dimethacrylate (GPDM). GPDM is more hydrophilic than 10-MDP, with two polymerizable groups and a shorter spacer chain. It may act similarly as HEMA since it is not able to form stable calcium salts like 10-MDP but can efficiently diffuse into the collagen network, rapidly penetrating it. Yoshihara et al., (2018) attributed the excellent bond performance of OBFL mostly to this monomer [36]. The ratio of acidic monomers has to be well optimised as the concentration has to be enough to guarantee initial demineralisation and chemical bonding but also low enough to prevent exaggerated hydrophilicity in the set material [37]. The second acidic monomer in OBFL, MEPP, is estimated to be around 15 to 20% in OBFL’s primer based on the information from the safety datasheet.

Solvents in primers are often a co-mixture of water/ethanol or water/acetone. Water is used as a co-solvent to improve hydrogen bonding and to take part in the ionisation of acidic monomers in SE or universal bonding systems. Water-based systems, such as CFSE, are rare compared to other formulations [2]. Their lower volatility makes them more difficult to dry. Consequently, residual water may be left in the monomers and subsequent polymer network that may affect polymerisation or cause plasticisation. Both of which can affect mechanical properties [4,38]. Ethanol, in OBFL, is volatile and much easier to evaporate. It may also enable a reduction in water content thereby improving stabilisation of the adhesive interface, which is commonly subject to hydrolytic degradation. This study suggests CFSE contains four times the amount of water of OBFL. High water content, acid ionisation, and acid content may increase the depth of hybrid layer formation and etching in SE systems. The water level must therefore be carefully optimised. Higher amount of solvents result in residual solvent remaining, which makes adhesive layers thin and can lead to the creation of pores in the interface, with poor outcome [39].

Primer application in this study, by the 2-step SE and 3-step ER systems, was succeeded by a hydrophobic bonding resin (adhesive). The adhesive increases the film thickness and renders the interface impermeable, preventing the effect of hydrolysis, retarding enzymatic degradation [40].

Using an ATR-FTIR technique to assess chemical composition has many benefits since it is an easy technique to use, especially when analysing liquids such as bonding systems. It is sensitive enough to identify different chemical groups but can be particularly useful to monitor changes during reactions, using continuous spectral acquisition to understand which chemical groups react [18,22,23]. It does however have some limitations. Low level components are difficult to identify, since a mixture of various monomers contains several chemical groups that can overlap in key absorption regions, complicating identification. Interaction of monomers in the mixture occurs, leading to peaks shifting. Due to this, a model prediction of the individual components may not perfectly fit the spectrum of the pure mixture. This justifies some of the differences seen in this study. Strong peaks can also swamp other important peaks, examples of this are the water absorption peak at 1635 cm^−1^ which competes with the C=C aliphatic stretch in the same region and the glass region in mixtures with filler particles (1280–800 cm^−1^) [18]. However, the presence of UV scavengers, inhibitors, and initiators will not cause peak shifts as the levels are far too low to appear in the infra-red spectra. Filler absorption is related to the contact that fillers make with the diamond in the ATR system. This is determined by the level of pressure applied using the ATR golden gate and particle size and shape. Smaller particles provide strong spectra more readily. Conversely, liquid ATR-FTIR are considerably more reproducible. Quantification of filler load in the adhesives using FTIR is therefore not be possible.

There is a vast array of commercial dental bonding systems but studies quantifying their components are seldom found in the literature [2]. Thus, an estimation of the levels of the components can only be found in information supplied by the manufacturers, which is limited, or in safety datasheets that contain only certain components, leaving out others. An example of this is the SDS of OBFL’s adhesive which does not mention the presence of Bis-GMA, identifiable in this study, or the filler load included in the adhesive of CFSE, let alone the amount of solvent included in both formulations. Such omissions prevent clinicians from adjusting factors such as drying times or taking decisions regarding the polymerisation of their bonding systems in situ. Furthermore, components, even in low amounts should be disclosed together with, at least, a range of their relative amounts. Monomers, initiator systems, or other active ingredients may elicit toxic or allergic responses in surrounding tissues, raising the concern of safety in patients, if the composition is unknown to the clinician.

Since many methacrylates have similar spectra, a detailed examination is required to identify differences in bonding system chemistries. Shifts in strong C=O and C–O peaks whose wavenumbers are strongly affected by adjacent chemical groups can help to identify different monomers. The C=O stretching vibration (variation from 1700–1720cm^−1^) is also sensitive to intermolecular H-bonds with groups such as hydroxyl or phosphate groups on other molecules. This can explain the broadening or splitting of the C=O peak for HEMA, Bis-GMA, and 10-MDP but sharper peak in TEGDMA with the peak at lower wavenumbers being due to a hydrogen bonded carbonyl. Varying ionisation of carboxylic acid groups or its attachment to conjugated double bonds in aromatic rings in monomers such as MMEP can also have strong effects on the C=O and C–O peaks causing them to shift. Considering individual reference peaks, aromatic rings in Bis-GMA give distinctive peaks between 1500 and 1600 cm^−1^. Monomers that contain long hydrocarbon chains show weak methacrylate peaks at around 1450 cm^−1^ (C-H bond). The phosphate group in monomers such as 10-MDP and GPDM gives a broad strong peak around 1000 cm^−1^, however, due to the absorption of filler particles in adhesives in the same region it can become swamped.

Regarding filler load, bonding systems can be unfilled or can range from 0.5 wt.% to >40 wt.% [41]. OBFL is reported to have the highest filler content out of all the bonding systems in the market. This study confirms its 48 wt.% of filler content, which is higher than CFSE. This high filler load strengthens the hybrid layer and may act as an artificially elastic cavity wall, which may also explain its high bond strengths. TEM images confirm resin tags with a high amount of filler particles in dentine tubules, when OBFL is used [40]. This contributes to a uniform thickness in the formation of the hybrid layer, compared to other systems, linked to better mechanical properties [28,42]. For this reason, OBFL is consistently used in indirect restorations for luting/bonding purposes. OBFL has two types of fillers: Barium aluminosilicate and silica nanoparticles whilst it is common to find only silica particles in other bonding systems. As for CFSE, only one type of filler was found and in a much lesser quantity (5 wt.%). Differences in film thickness in both systems will be due to filler content. Both manufacturers recommend the application of only one coat of primer and adhesive.

During free-radical polymerisation carbon-carbon double bonds (C=C) are replaced by two C–C single bonds. When this occurs, the methacrylate carbonyl (C=O) group is also no longer conjugated to a C=C bond. This causes a change in its frequency of vibration and a shift in its associated peak to higher wavenumbers. Additionally, the C–O stretch doublet shifts to lower wavenumbers [22,23,43]. The difference spectra observed above upon polymerisation as a result of these changes are identical to previously published work on polymerising methacrylate based dental composites. This shows that the peak shifts that occur due to the polymerisation reaction are transversal to all light-curable resin-based dental materials, and these can be used to monitor levels of polymerisation [23].

The conversion rate is affected by a number of factors directly linked with the chemical structure and physicomechanical properties of monomers. In Bis-GMA-rich adhesive formulation, the *D_C_* (%) is generally limited due a lower flexibility of the monomer leading to a higher glass transition temperature (Tg). When monomers polymerise, their flexibility decreases and the polymer/monomer mixture glass transition temperature gradually rises. When this transition temperature reaches that of the surroundings, the system solidifies, and the polymerisation reaction halts. Molecules such as HEMA and TEGDMA with greater flexibility have lower Tg and therefore reach higher conversions before vitrification [10,44,45]. Steric hindrance also impairs dimethacrylate monomers such as Bis-GMA from completing the polymerisation reaction, owing to a restricted rotational ability of the aromatic rings in its structure [46]. *D_C_* (%) is also affected by the presence of acidic monomers, with lower values being reported in formulations containing monomers with acid groups. OBFL, containing two acidic functionalised monomers registered lower conversion levels [47]. The *D_C_* (%) obtained for the bonding systems in this study proves that this conversion is predominantly material and compositional-dependent, since significant differences were found between the systems. Sato et al. (2017) described a newly added photo-initiator in CFSE 2, which is responsible for increasing the degree of conversion when compared to its predecessor, CFSE [48]. This could explain its better performance compared to OBFL. Differences in filler loads can also be responsible for differences in *D_C_* (%). Higher filler loads are related to conformational restrictions on the molecules, caused by the filler surface, which in turn slows down reactions [49]. CFSE conversion values reported in this study are higher than the ones found in the literature, even though studies evaluating this novel bonding system are scarce [48,50]. ATR-FTIR is thus a quick and relatively simple method to monitor how fast polymerisation occurs in bonding systems, giving useful information as to how long these materials should be cured for, in order to achieve minimum conversion levels. The ATR-FTIR technique is also useful to investigating the depth of cure by increasing sample thickness [51]. In resin composites, scattering by the fillers, as well as absorbance by the initiator is important in limiting lower reaction. Photobleaching of the initiator is required to enable light penetration at depth. With bonding systems, as fillers are in lower percentages, this absorbance has higher effect than scattering.

Solvent evaporation during the bonding procedure has to be thorough in order to achieve an acceptable conversion, and the evaporation rate depends upon the solvent [52]. Simplified systems with a high amount of hydrophilic monomers and solvent generally achieve suboptimal polymerisation. Entrapped water is also responsible for nanoleakage and hybrid layer degradation [30]. Conversely, hydrophobic and solvent-free primers reveal less permeable interfaces and a more complete setting reaction [11]. A thorough solvent evaporation increases monomer concentration and allows monomers that are far away from each other to reduce their spatial distance so they can react during polymerisation. This also prevents the softening of the polymer due to the residual presence of water, acetone, or ethanol which affects cross-linking and tensile strength [11,16]. CFSE is recommended to be dried for more than 5 s by the manufacturer, while for OBFL, 5 s is the recommended time. In this study, 20 s was the drying time, which may mean conversion rates achieved clinically can be substantially lower. Although a higher amount of primer/adhesive may have been used in this study than it is in certain clinical scenarios, drying times and polymerisation is required to compensate for this. For water containing formulations, at least 15 s is recommended [40]. A fast polymerisation and conversion rate allows polymers to reach their final mechanical properties, ensuring that bonding systems form less permeable, stronger, and more stable hybrid layers.

All of these factors have an impact on the performance of bonding systems and knowledge of components allows clinicians to comply to sensitivity, choice, and clinical applicability.

## 5. Conclusions

OBFL and CFSE, both gold standard bonding systems, have differences in component ratios, filler load, and type. The ATR-FTIR technique proved to be a useful method to help identify key components and study changes that occur due to chemical reactions, such as polymerisation. According to modelling, OBFL was found to contain a 30% water/ethanol co-mixture vs. 60% water for CFSE. Bis-GMA/HEMA mixtures and acidic-functionalised molecules were present in both adhesive strategies. FTIR modelling provided a rapid evaluation of the initial composition of bonding systems. OBFL was found to have the highest filler load (48% vs. 5%), with two different types of fillers. Final degrees of conversion were also variable, with CFSE reporting higher *D_C_* (%) than OBFL (79 ± 2% vs. 74 ± 1%). The systems showed high conversion rates within seconds of light exposure. This technique allowed a monitorisation of how fast the reaction goes versus final conversions and changes associated to the reaction.

## Figures and Tables

**Figure 1 materials-14-00760-f001:**
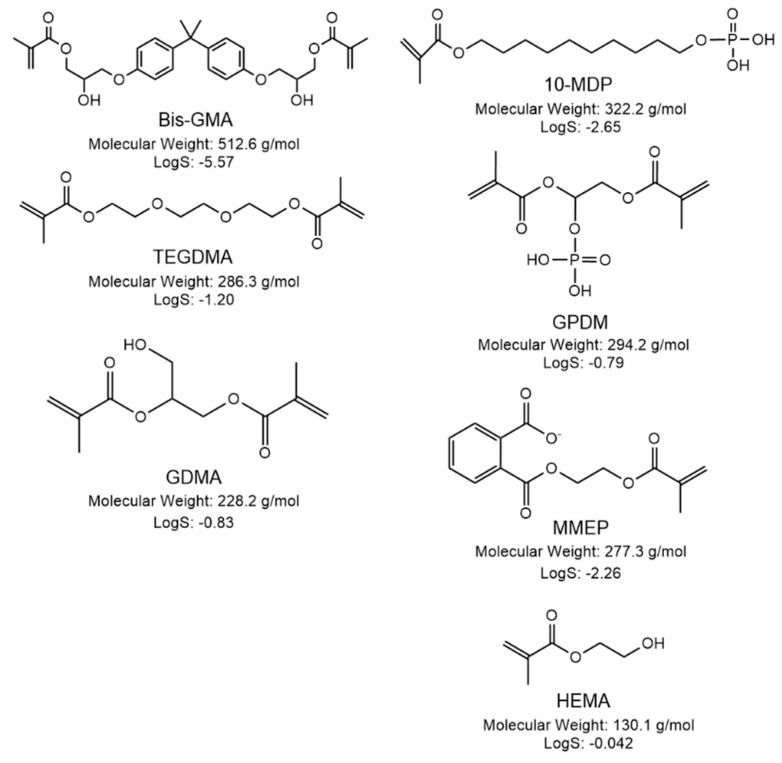
Chemical structure and properties (molecular weight and solubility in water—LogS) of monomers used in the bonding systems.

**Figure 2 materials-14-00760-f002:**
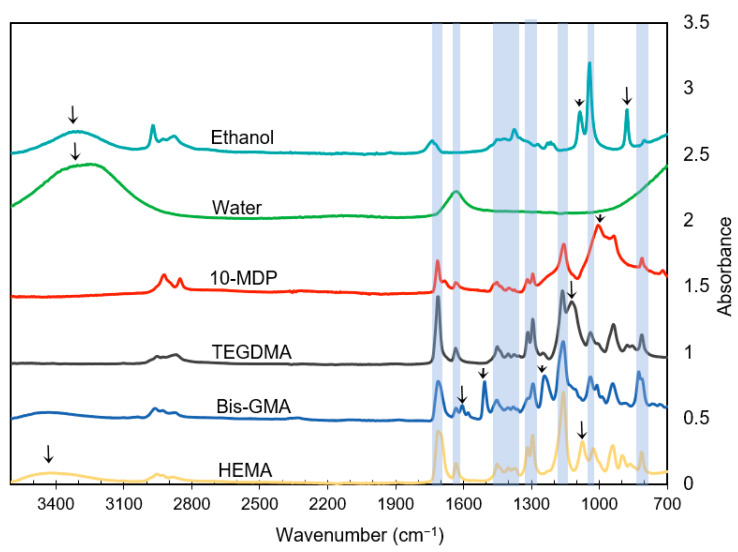
FTIR spectra of dental monomers and solvents used in bonding systems (3600–700 cm^−1^). Arrows indicate strong, well-separated peaks that can be used to identify components. These are not overlapped by common methacrylate peaks which are highlighted by blue bands.

**Figure 3 materials-14-00760-f003:**
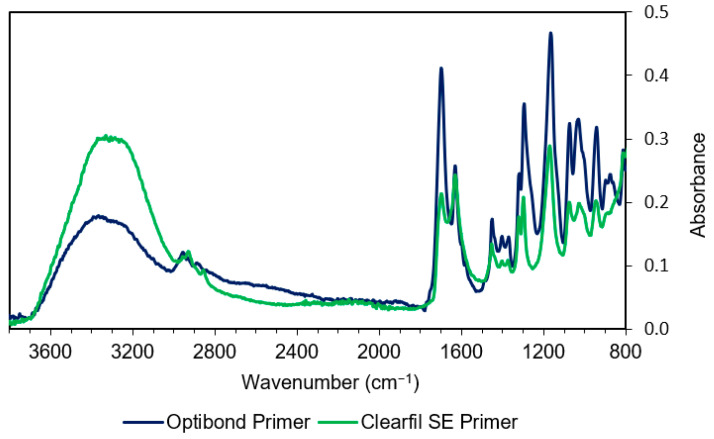
Comparison of IR spectra of both primers. Note the higher absorbance in the –OH region (3300 cm^−1^) and lower methacrylate peaks at 1700, 1320, 1300, 1160, 1080, 1050, and 940 cm^−1^ with CFSE due to its greater water content.

**Figure 4 materials-14-00760-f004:**
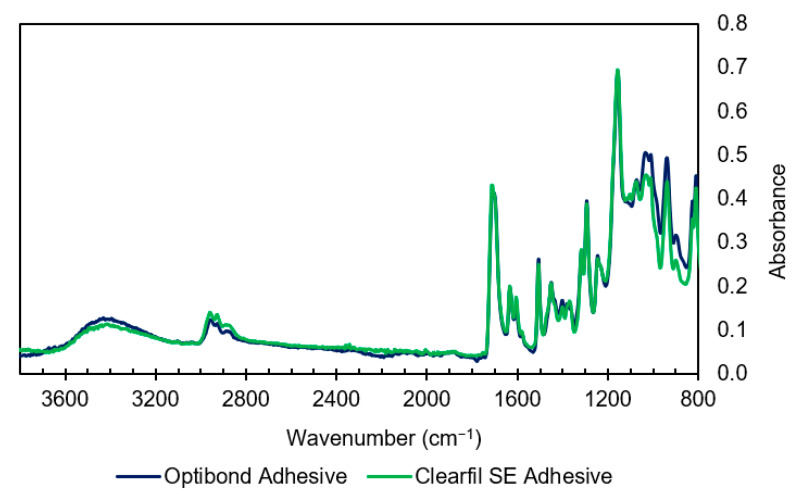
Comparison of IR spectra of both adhesives. The mixtures appear to be similar except in the 1000–900 cm^−1^ region which can be due to differences in filler load and also the fraction of monomers such as HEMA and GDMA.

**Figure 5 materials-14-00760-f005:**
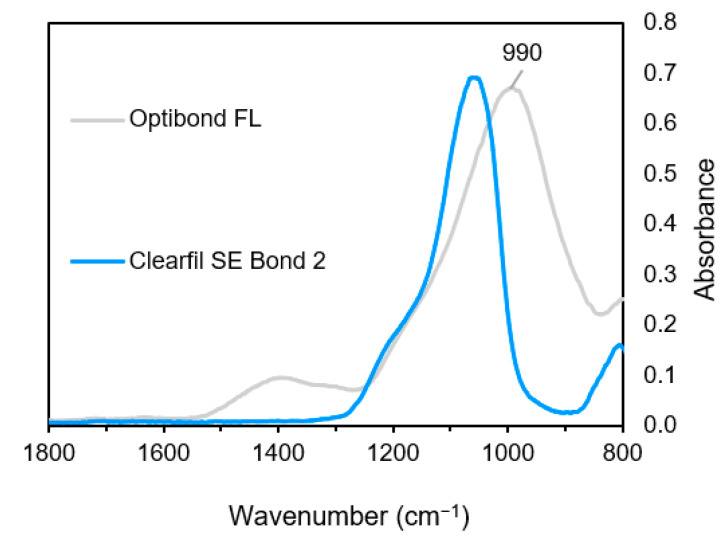
FTIR spectra of fillers retrieved from the adhesives (1800–800 cm^−1^).

**Figure 6 materials-14-00760-f006:**
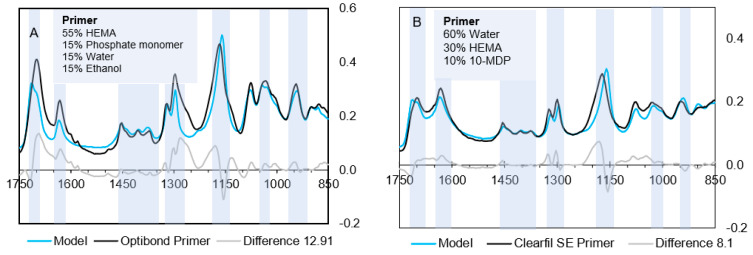
FTIR spectra of primers, the calculated model spectra, percentages of pure spectra used in the model, and difference between the actual spectra and model. (**A**) OBFL primer model requires a mixture of water and ethanol and a higher amount of HEMA. (**B**) The CFSE model requires a higher percentage of water. Blue bars highlighted represent common methacrylate peaks.

**Figure 7 materials-14-00760-f007:**
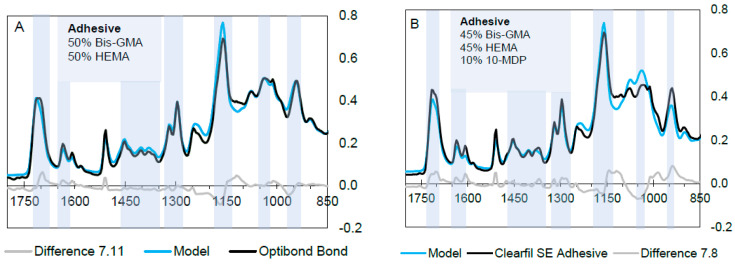
FTIR spectra of primers, the calculated model spectra, and difference between the actual spectra and model. OBFL adhesive (**A**) contains 50% Bis-GMA and 50% HEMA. Level of GDMA was too low to detect. CFSE’s adhesive (**B**) right figure, the model has an equal mixture of Bis-GMA/HEMA. Blue lines highlighted represent common methacrylate peaks.

**Figure 8 materials-14-00760-f008:**
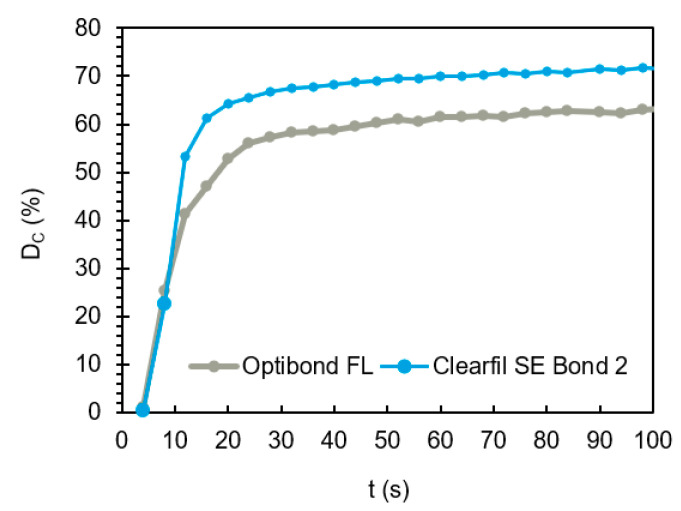
*D_C_* (%) versus time for mixed OBFL and CFSE bonding systems upon 20 s light exposure. At 10 s, the systems reach or surpass 50% of the maximum conversion level, showing that the polymerisation reaction is fast once light irradiation begins.

**Figure 9 materials-14-00760-f009:**
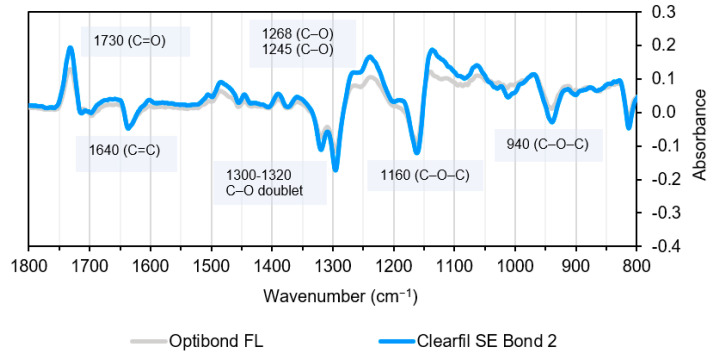
Absorbance changes for OBFL and CFSE, after polymerisation. Note shift of C=O stretch peak around 1700 cm^−1^ to 1730 cm^−1^; loss of a 1635–1640 cm^−1^ C=C stretch peak and shift of 1300–1320 cm^−1^ C–O stretch doublet to lower wavenumbers at 1268 and 1245 cm^−1^.

**Table 1 materials-14-00760-t001:** Bonding system source, type and composition according to the manufacturers and safety.

Material	Type	Primer	Adhesive
Optibond™ FL (Kerr, USA) **OBFL**	3-step ER bonding system	HEMA (10–30%) MMEP (10–30%) GPDM (5–10%) Solvent: Water/Ethanol	HEMA (10–30%) Bis-GMA (N/A%) GDMA * Filler: Barium aluminosilicate Sodium hexafluorosilicate and fumed silica with silane (48 wt.%)
Clearfil SE 2 (Kuraray, JP) **CFSE**	2-step SE bonding system	HEMA (20–40%) 10-MDP * Hydrophilic aliphatic Dimethacrylate * Solvent: Water	HEMA (20–40%) Bis-GMA (25–45%) 10-MDP * Hydrophobic aliphatic Dimethacrylate * Filler: Colloidal silica (N/A%)

Percentages shown according to SDS (safety datasheets). * indicates low level. Both formulations also contain low levels of camphorquinone light activated initiator. 10-MDP: 10-methacryloyloxydecyl dihydrogen phosphate; Bis-GMA: Bisphenol-A-glycidyl dimethacrylate; GDMA: Glycidyl dimethacrylate; GPDM: Glycerophosphate dimethacrylate; HEMA: Hydroxy ethyl methacrylate; N/A: Not available in the information supplied; MMEP: Mono(2-methacryloyloxy) ethyl phthalate.

**Table 2 materials-14-00760-t002:** IR assignments for common dental monomers and solvents—left side peaks are due to the methacrylate group and right side peaks are due to solvents and groups attached to the methacrylate that may be used to narrow down which methacrylates are present.

Wavenumber (cm^−1^)	Methacrylate Assignment	Wavenumber (cm^−1^)	Assignment	Compound
2940	C–H stretch	3400	O–H stretch	HEMA, Bis-GMA
1700–1720	C=O stretch	3300	O–H stretch	Water, Ethanol
1640	C=C stretch	1635	O-H bend	Water
1350–1450	C–H bending	1610	Aromatic C=C	Bis-GMA
1320, 1300	C–O stretch doublet	1510	Aromatic C=C	Bis-GMA
		1240	Aromatic C–O	Bis-GMA
		1120	C–O–C stretch	TEGDMA
		1080	C–OH stretch	HEMA
		1050/1090	C–OH stretch	Ethanol
		1000	P–O stretch	10-MDP
		900	C–C–O stretch	HEMA
		880 830	C–C–O stretch C–C–O stretch	Ethanol Bis-GMA
		650–900	Water hindered rotation	Water

**Table 3 materials-14-00760-t003:** IR assignments for R groups of monomers in OBFL and CFSE primers and adhesives.

Wavenumber (cm^−1^)	Assignment	Compound	Bonding System
1635	O–H stretch, Aliphatic C=C	Water, Monomers	OBFL, CFSE
1604–1610	Aromatic C=C	Bis-GMA, MEPP	OBFL, CFSE
1510	Aromatic C=C	Bis-GMA, MEPP	OBFL, CFSE
1240	Aromatic C–O	Bis-GMA	OBFL, CFSE
1080	C–OH stretch	HEMA	OBFL, CFSE
1050/1090	C–OH stretch	Ethanol	OBFL
1000–1010	P–O stretch/Si–O	10-MDP GPDM	CFSE OBFL
900	C–C–O stretch	HEMA	OBFL, CFSE

**Table 4 materials-14-00760-t004:** Summary table indicating the fractions of different spectra (X in Equation (1)) used to produce the model spectra for OBFL and CFSE. Also provided is the required flat background term and the sum of the modulus of differences that indicates the quality of model fit (Equation (4)).

Components	Pure Spectrum Fraction (X) Primer		Adhesive	
	OBFL	CFSE	OBFL	CFSE
**HEMA**	0.55	0.30	0.50	0.45
**Bis-GMA**	-	-	0.50	0.45
**10-MDP**	0.15	0.10	-	0.10
**Water**	0.15	0.60	-	-
**Ethanol**	0.15	-	-	-
**Filler**	-	-	0.32	0.30
**Background absorbance**	0.04	0.02	0.02	0.03
**Sum of mod (difference)**	12.9	8.1	7.3	7.8

**Table 5 materials-14-00760-t005:** Means and standard deviation of filler load and polymerisation kinetics: *D_C_* (%) = final extrapolated degree of conversion and Rp (%s^−1^) is the maximum rate of reaction (*n* = 3). Different capital letters in the same column that indicate difference is statistically significant at (*p* < 0.05). *D_C_* (%) was analysed using a T-Test, while R_p,max_ was analysed with a Mann–Whitney U test.

Material	Filler (wt.%)	*D_C_* (%)	R_p,max_ (%s^−1^)
**OBFL**	48 ± 3	74 ± 1 ^A^	4.0 ± 1.6 ^A^
**CFSE**	5 ± 1	79 ± 2 ^B^	5.0 ± 0.1 ^A^

**Table 6 materials-14-00760-t006:** Table comparing fractions found in this study to fractions supplied in the safety datasheets and information derived from suppliers (retrieved from Table 1). Y (yes) indicates the fractions were within the range reported by the manufacturer, N (no) indicates they fell outside the range, while N/A (not available) was added when there was no information present.

Components		Component Fractions (FTIR and Filler Load Determination)						
	OBFL P	Kerr	OBFL A	Kerr	CFSE P	Kuraray	CFSE A	Kuraray
**HEMA**	0.55	N ^1^	0.50	N	0.30	Y	0.45	N
**Bis-GMA**	-	-	0.50	N/A	-	-	0.45	Y
**GPDM/10-MDP**	0.15	N ^1^	-	-	0.10	N/A	0.10	N
**Water**	0.15	N/A	-	-	0.60	N/A	-	-
**Ethanol**	0.15	N/A	-	-	-	-	-	-
**Filler**	-	-	0.48	Y	-	-	0.05	N/A

(P) represents primer, while (A) represent adhesive. ^1^ The model might be taking into account spectral representations of both GPDM and MMEP in both HEMA and 10-MDP. 10-MDP was modelled as a substitution for GPDM.

## Data Availability

The data is available upon request to the corresponding author.

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
