# Peer review of "Modelling ATR-FTIR Spectra of Dental Bonding Systems to Investigate Composition and Polymerisation Kinetics"

_materials, 2021, doi:10.3390/ma14040760_

Round 1

Reviewer 1 Report

The paper is well written and focus a area of interest. The authors did a complete study using FTIR technology, that can help other authors to plan their work. The paper does not present a new product, however is innovative with the methodology used to study two commercial products. I enjoyed read this paper.

As suggestion, authors may consider the elaboration of guidelines to perform such measurements, not only in dental materials, but for other applications of interest. The development of a methodology can be very interesting for scientific and medical/pharmaceutical community.

Regarding the paper, I will propose some revisions and some notes that can improve the paper: 

Line 53 - the first time that GPDM and 10-MDP acronyms appear should be accompanied by its full name.

Line 94 - TEGCMA, should appear with full name.

Line 325 – Figure 7 A and B is overlapped in the given pdf. Surely this issue will be corrected, but it was hard to evaluate this figure.

Line 327 – the description of CFSE´s adhesive is incomplete, authors should mention the 10% of 10-MDP, present in the figure itself. The way it is written seems that CFSE is the same as OBFL adhesive.

Line 544 – The increasing in the study drying time, compared to the recommended clinically time, can also be related to the sample amount, and therefore, more sample, more time to achieve conversion stability. Also, penetration of LCU light can be difficult with more amount of sample than the one used clinically, and therefore increasing the conversion time. I have not enough information to verify this, but authors can say if this makes sense.

Line 552 - At the end authors should present a table similar to table1, where they present the percentages founded by then with this study, and compare it with the commercial label. Also authors can elaborate the implications on clinical safety of such a different composition in these materials, especially in the ones where differences were more clear (solvent type and amount, filler amount, etc.)

Author Response

Overall, we thank the reviewer for the pertinent suggestions and revision points, which have contributed to enhance the scientific reporting and quality of our paper.

Point 1: Line 53 - the first time that GPDM and 10-MDP acronyms appear should be accompanied by its full name.

Response: We have now corrected this so the acronyms are explained in full (please see line 53)

Point 2: Line 94 - TEGDMA, should appear with full name.

Response: We have now corrected this, and the full name of TEGDMA is shown (please see line 95)

Point 3: Line 325 – Figure 7 A and B is overlapped in the given pdf. Surely this issue will be corrected, but it was hard to evaluate this figure.

Response: The second reviewer also pointed this out. We have inserted page breaks, although this error only happens after the docx is submitted onto the platform and may only be able to be solved by the editorial team.

Point 4: Line 327 – the description of CFSE´s adhesive is incomplete, authors should mention the 10% of 10-MDP, present in the figure itself. The way it is written seems that CFSE is the same as OBFL adhesive.

Response: We agree with the reviewer and also believe that readers may think the composition is the same. We have included a sentence which clearly explains this is a difference between both materials, and that the adhesive of CFSE contains 10% 10-MDP (please see line 335).

Point 5: Line 544 – The increasing in the study drying time, compared to the recommended clinically time, can also be related to the sample amount, and therefore, more sample, more time to achieve conversion stability. Also, penetration of LCU light can be difficult with more amount of sample than the one used clinically, and therefore increasing the conversion time. I have not enough information to verify this, but authors can say if this makes sense.

Response: We thank the reviewer for bringing out a valid point. We have now included additional sentences addressing this compensation and explaining it. Indeed, the amount of sample influences the drying time and polymerisation properties and should be adjusted (please see line 597)

Point 6: Line 552 - At the end authors should present a table similar to table1, where they present the percentages founded by then with this study and compare it with the commercial label. Also, authors can elaborate the implications on clinical safety of such a different composition in these materials, especially in the ones where differences were more clear (solvent type and amount, filler amount, etc.)

Response: Again, we thank the reviewer for the pertinent suggestion. Due to this we have added a new section (3.3) with comparisons to the information supplied by the manufacturers. To link with this new set of results, we have also expanded on the implications of omitting component ratios and what this means for the clinician and for the patient, please see the 10th paragraph (Discussion)

Reviewer 2 Report

The paper “Modelling ATR-FTIR spectra of dental bonding systems to investigate composition and polymerisation kinetics” submitted for review in Materials deals with the investigation on the modeling of FTIR spectra of dental bonding systems by correlating spectra of individual components and mixtures to study commercially available compositions. Comparison of the intensities of individual spectral bands was also used to study the polymerization kinetics of these compositions.

In addition to describing the study and results, the authors included in the discussion a brief description and characterization of the standard components of commercially available compositions of dental bonding systems.

The article is relatively well written without significant errors, but some points need to be corrected or clarified for better understanding of the content by the reader:

  • Line 13 and 14 - the meaning of n=3 should be briefly explained
  • abbreviations used in the text for the first time should be explained in the same place
  • Line 227 – “both primer have a strong OH peak at 1080 cm-1 “ – In the spectrum of Optibond Primer “blue line” peak at 1080 cm-1 is stronger
  • Figure 7 has to be corrected - overlaying spectra and text in this way is unreadable and must be corrected
  • Line 320 - the 1200 - 750 spectral range is discussed - probably the 1200 - 1750 cm-1 range was meant
  • Table 5 - indexes A and B should be clearly described
  • Line 442 – “ATR-FTIR technology” - ATR-FTIR is a technique or method rather than a technology.
  • The authors do not state whether all spectra were taken using the ATR technique and whether ATR spectra correction was used (e.g., fig. 2, fig. 4, etc.).
  • The authors do not state the thickness of the component layers tested. When recording the spectra, it is important to obtain a layer thickness of at least about 20 micrometers - the maximum penetration of the material by the IR beam.
  • When studying kinetics and the degree of conversion, the reaction must occur uniformly throughout the layer for the results to be reasonable. The composition should be transparent enough to the rays that initiate polymerization that the conversion is the same in the boundary layer analyzed by the spectrometer as it is at the irradiated surface.

It seems that the method described to study the composition and kinetics of polymerization is very interesting, it is certain that more accurate techniques exist to determine the composition of such mixtures. However, they are certainly more time consuming and more expensive.

I think that after correcting the described points the article can be published in Materials.

Author Response

Overall, we thank the reviewer for the pertinent suggestions and revision points, which have contributed to enhance the scientific reporting and quality of our paper.

Point 1: Line 13 and 14 - the meaning of n=3 should be briefly explained; abbreviations used in the text for the first time should be explained in the same place

Response: We have now changed this in accordance to what was suggested by the reviewer as we explained we undertook 3 repetitions. Abbreviations were now defined in the text as well, please see changes in lines 13 and 13

Point 2: Line 227 – “both primer have a strong OH peak at 1080 cm-1“ – In the spectrum of Optibond Primer “blue line” peak at 1080 cm-1 is stronger

Response: We have now made this clear, however, we did intend to say that the peaks are strong – they are always compared relatively to absorbance of other peaks in the same spectrum. For CFSE, the absorbance of all peaks in that region is lower due to high water content. Relative to other peaks in the fingerprint region, the OH peak in CFSE can still be considered a strong and distinctive peak. Please see lines 240-243.

Point 3: Figure 7 has to be corrected - overlaying spectra and text in this way is unreadable and must be corrected

Response: This was a point also addressed in the comments of the first reviewer. We have now inserted page breaks in the correct places, although this issue may only be fixable by the editorial office – because when we save it as a pdf it always shows figures correctly.

Point 4: Line 320 - the 1200 - 750 spectral range is discussed - probably the 1200 - 1750 cm-1 range was meant

Response: We thank the reviewer for the careful attention he took in reading all details, although in this case we do mean to discuss the 1200-750 cm-1, as this is the wavenumber range where chemical bonds and functional groups of glass fillers absorb light in the IR region.

Point 5: Table 5 - indexes A and B should be clearly described

Response: The reviewer raised a pertinent comment, and the caption has now been fully revised to make the details of the differences clearer. See lines 356-358.

Point 6: Line 442 – “ATR-FTIR technology” - ATR-FTIR is a technique or method rather than a technology.

Response: The reviewer is right in point out this fault, which has now been corrected in line 481.

Point 7: The authors do not state whether all spectra were taken using the ATR technique and whether ATR spectra correction was used (e.g., fig. 2, fig. 4, etc.).

Response: We have now included the correct information for this, please see line 125.

Point 8: The authors do not state the thickness of the component layers tested. When recording the spectra, it is important to obtain a layer thickness of at least about 20 micrometers - the maximum penetration of the material by the IR beam.

Response: We thank the reviewer for bringing this up, and we have now confirmed that the thickness was larger than this value and provided quantitative data in lines 190-194.

Point 9: When studying kinetics and the degree of conversion, the reaction must occur uniformly throughout the layer for the results to be reasonable. The composition should be transparent enough to the rays that initiate polymerization that the conversion is the same in the boundary layer analyzed by the spectrometer as it is at the irradiated surface.

Response: We also thank the reviewer, as we have addressed this question in lines 190-194 by justifying the standardization and depth of light cure, as well as in the discussion (please see lines 579-584.
